# Comparison of Multiple Carbapenemase Tests Based on an Unbiased Colony-Selection Method

**DOI:** 10.3390/biomedicines12092134

**Published:** 2024-09-20

**Authors:** Hsin-Yao Wang, Yi-Ju Tseng, Wan-Ying Lin, Yu-Chiang Wang, Ting-Wei Lin, Jen-Fu Hsu, Marie Yung-Chen Wu, Chiu-Hsiang Wu, Sriram Kalpana, Jang-Jih Lu

**Affiliations:** 1Department of Laboratory Medicine, Chang Gung Memorial Hospital at Linkou, Taoyuan 333, Taiwan; mdhsinyaowang@gmail.com (H.-Y.W.); wal018@health.ucsd.edu (W.-Y.L.); weitinglin66@gmail.com (T.-W.L.); ab668899@cgmh.org.tw (C.-H.W.); vbkk2005@gmail.com (S.K.); 2School of Medicine, National Tsing Hua University, Hsinchu 300044, Taiwan; 320/20 GeneSystems, Gaithersburg, MD 20877, USA; 4Department of Computer Science, National Yang Ming Chiao Tung University, Hsinchu 112304, Taiwan; yjtseng@nycu.edu.tw; 5Computational Health Informatics Program, Boston Children’s Hospital, Boston, MA 02115, USA; 6Department of Medicine, University of California San Diego, San Diego, CA 92093, USA; 7Department of Medicine, Brigham and Women’s Hospital, Boston, MA 02115, USA; 8Department of Medicine, Harvard Medical School, Boston, MA 02115, USA; 9Division of Pediatric Neonatology, Department of Pediatrics, Chang Gung Memorial Hospital, Linkou, Taoyuan 33305, Taiwan; hsujanfu@cgmh.org.tw; 10Department of Medicine, MetroWest Medical Center, Framingham, MA 01702, USA; 11Department of Laboratory Medicine, Keelung Chang Gung Memorial Hospital, Keelung 204201, Taiwan; 12Division of Clinical Pathology, Taipei Tzu Chi Hospital, Buddhist Tzu Chi Medical Foundation, New Taipei City 23142, Taiwan

**Keywords:** carbapenemase, unbiased colony selection, mCIM/eCIM, Carba5, CPO panel

## Abstract

Carbapenemase-producing organisms (CPOs) present a major threat to public health, demanding precise diagnostic techniques for their detection. Discrepancies among the CPO tests have raised concerns, partly due to limitations in detecting bacterial diversity within host specimens. We explored the impact of an unbiased colony selection on carbapenemase testing and assessed its relevance to various tests. Using the FirstAll method for unbiased colony selection to reduce bias, we compared the results from different methods, namely the modified carbapenem inactivation method/EDTA-modified carbapenem inactivation method (mCIM/eCIM), the Carba5, the CPO panel, and the multiplex PCR (MPCR). We compared the FirstAll method to the conventional colony selection for MPCR with seven CPO species. In addition, we evaluated the test performance on seven CPO species using MPCR as a reference and the FirstAll method as the colony-selection method. The results revealed that the selections from the FirstAll method have improved rates of carbapenemase detection, in comparison to approximately 11.2% of the CPO isolates that were noted to be false negatives in the conventional colony-selection methods. Both the Carba5 test and the CPO panel showed suboptimal performance (sensitivity/specificity: Carba5 74.6%/89.5%, CPO panel 77.2%/74.4%) in comparison to the FirstAll method. The Carba5 test provided specific carbapenemase class assignments, but the CPO panel failed in 18.7% of the cases. The Carba5 test and the CPO panel results correlated well with ceftazidime–avibactam minimal inhibitory concentrations (MICs). The concordance for Class A/D with MICs was 94.7% for Carba5 and 92.7% for the CPO panel; whereas for Class B, it was 86.5% for Carba5 and 75.9% for the CPO panel. In conclusion, FirstAll, as the unbiased colony-selection method, was shown to impact carbapenemase testing. With FirstAll, the diagnostic performance of both the Carba5 and the CPO panel was found to be lower. Furthermore, the utilization of ceftazidime–avibactam guided by either the CPO panel or Carba5 was appropriate.

## 1. Introduction

Carbapenemases are β-lactamase enzymes that break down carbapenem antibiotics, and they belong to molecular Class A, B, and D beta-lactamases. Carbapenems are effective antibiotics that are used in the treatment of multidrug-resistant organisms [1]. Carbapenemase-producing (CP) organisms, especially the members in the *Enterobacteriaceae* family (CP-CRE), are a global public health concern [1], given the rapid increase in their prevalence [2]. The rapid increase in CP-CRE is a public health issue because the resistant genes can be transmitted via plasmids [3]. Carbapenemase mechanism testing is important for carbapenem-resistant *Enterobacteriaceae* (CRE) prevention because CP-CRE disseminates more readily [4] and with a poorer prognosis than non-CP-CREs [5], thus requiring an intensive infection control approach. The discrepancies between the different kinds of carbapenemase tests, such as modified carbapenem inactivation method (mCIM)/EDTA-modified carbapenem inactivation method (eCIM), Carba5, and molecular tests, arise from the underlying mechanisms [6,7] and the variations in the diagnostic accuracies. Genetic tests are more sensitive, whereas phenotypic tests are only relevant for the functionality [8]. Currently, with the low degree of automation for tests, the inter-operator variations also contribute to variations in results [9]. In addition, selection bias results in picking colonies from an isolate with heterogeneity leading to discrepancy.

In clinical microbiology laboratories, typically, bacteria of the same species are assumed to be homogeneous [10]. However, it becomes complicated in patients with multiple medical conditions; underlying immunosuppression, the evolution of bacteria due to a history of infections is induced by longer or repeated exposure to antibiotics treatment [11]. Multiple factors impose stress on bacteria that alter the interaction between microorganisms and hosts. In response to novel stresses, bacteria evolve using a variety of strategies to survive in vivo [12], thus transforming a relatively homogenous bacterial colony to a heterogenous one that is no longer genetically identical after this process. Theoretically, bacterial heterogeneity is an overlooked reality that is worthy of detailed investigation.

Bacterial heterogeneity has raised considerable attention in recent years. The genetic and phenotypic diversity could be identified between the colonies in carbapenem-resistant *Klebsiella pneumoniae* (CRKP) bloodstream infections (BSIs) [13]. The heterogeneous strains also exhibited significant differences in virulence in animal models. The data have suggested a new paradigm of CRKP population diversity causing BSIs, which challenges the single-organism hypothesis and has implications for understanding antibiotic resistance and pathogenesis further. Culture-based detection methods are time-consuming, along with limited intra-sample abundance, strain diversity information, and uncertain sensitivity [14].

Similarly, in a study of enteric vancomycin-resistant *Enterococcus* (VRE), clonal diversities and dynamic changes were observed over several weeks [15]. These findings suggest that traditional typing methods that analyze one isolate per patient may be insufficient for outbreak surveillance of VRE in highly vulnerable patients. Together, these studies [13,14,15] highlight the importance of considering within-host microbial diversity and bacterial heterogeneity in the context of infectious control. They also challenge traditional assumptions about the causes and mechanisms of infectious diseases. Moreover, the data suggest that within-host bacterial heterogeneity could be the cause of discrepancies noted between different lab tests. New approaches to detection and surveillance are likely necessary. To increase the ability to detect the complex and everchanging bacterial populations within individual hosts, we propose a non-biased colony-collection method (i.e., FirstAll).

The aim of the study was to compare the results of carbapenemase detection by PCR in seven *Enterobacteriaceae* species using two different colony-collection methods—FirstAll and the conventional method—as well as to compare the performance of several carbapenemase detection tests using the FirstAll method for colony collection with seven *Enterobacteriaceae* species. Theoretically, we hypothesize that the FirstAll method would be more effective than the conventional colony-picking method for carbapenemase detection. Based on this hypothesis, we evaluated the analytical performance of various carbapenemase tests using the FirstAll method. This study comprehensively analyzes the performance of various carbapenemase tests and contributes to our understanding of within-host bacterial heterogeneity and its impact on diagnostic methods.

## 2. Materials and Methods

### 2.1. Strain Re-Isolation and Species Identification

Seven potential CPO species—*Klebsiella pneumoniae* (*n* = 281), *Escherichia coli* (*n* = 59), *Enterobacter cloacae complex* (*n* = 59), *Klebsiella aerogenes* (*n* = 20), *Klebsiella oxytoca* (*n* = 17), *Citrobacter freundii* (*n* = 19), and *Citrobacter koseri* (*n* = 20)—were re-isolated from a bacterial bank. Selection bias was avoided by collecting all the bacterial lawns from agar plates. Isolates were stored at −70 °C until analysis. Fresh colonies grown on BBL™ Trypticase™ Soy Agar with 5% Sheep Blood (TSA II) (Becton, Dickinson and Company, Franklin Lakes, NJ, USA) for 24 h were picked and smeared onto a MALDI target plate (Bruker Daltonics GmbH & Co. KG, Bremen, Germany) in thin films for the identification of bacterial species as described earlier [16].

### 2.2. Conventional Colony and Unbiased Colony-Selection Methods

We proposed an unbiased bacterial colony-selecting method prior to further analyses. As mentioned above, we have stored the strains with an unbiased method (i.e., scratching all the bacterial lawns off from the agar plates). On this basis, we inoculated the storage onto agar plates and reconfirmed the bacterial species with MALDI-TOF. On the agar plates, furthermore, we adopted two different colony-selection methods, including the “conventional method” and the unbiased selection method (referred to as the “FirstAll method” (Figure 1A)). While performing the “conventional method”, several single colonies only were collected for further carbapenemase analyses. For the “FirstAll method”, we scratched over the first lawn on agar plates with 10 µL loops. The direction of collecting is vertical to the streaking of the first lawn. Thus, this collection method theoretically maximizes the collection of various colony variants. The FirstAll method was adopted as the colony-selection method for the carbapenemase tests.

The mCIM and eCIM tests were conducted on bacterial isolates as described by CLSI for *Enterobacterales* [6]. The test was performed in triplicate, and the results were interpreted by three independent technicians.

### 2.3. BD CPO Panel

The BD Phoenix CPO panel (BD, Franklin Lakes, NJ, USA) includes the ID/NMIC 504 test for detecting carbapenemase activity and the NMIC 500 test for Ambler classification. The CPO panel utilizes meropenem, doripenem, temocillin, and cloxacillin, alone and in combination with various chelators and beta-lactamase inhibitors for the detection and classification of CPOs. Specific Ambler classification was not provided for the cases with negative results. The test results were then interpreted using BD EpiCenter™ software V7.0 (BD) as per the latest CLSI or EUCAST guidelines.

### 2.4. Carba5 Test

For the CARBA-5 test (NG-Biotech, Guipry, France), five drops of the extraction buffer were mixed with a full 1 µL inoculation loop of CPO bacteria culture, and from this, 150 µL was transferred into the CARBA-5 cassette. The results were read after 15 min of incubation at room temperature.

### 2.5. Multiplex PCR (MPCR)

A multiplex PCR targeting important carbapenemase genes was designed (Table 1). In developing the MPCR, *K. pneumoniae* BAA-1705 and *K. pneumoniae* BAA-2146 strains were used as the KPC-positive and NDM-positive control strains. In addition to ATCC strains, three in-house strains were sequenced for their whole genome and served as the IMP-positive (*E. cloacae*), VIM-positive (*K. oxytoca*), and OXA-48-positive (*K. pneumoniae*) control strains. *K. pneumoniae* BAA-1706 was used as the carbapenemase-negative control strain.

DNA was extracted from isolated colonies with a QIAamp DNA mini kit (Qiagen, Taipei city, Taiwan). The concentration and purity of the extracted DNA were verified using a Nanodrop-ND1000 (Thermo Fisher Scientific, Waltham, MA, USA). PCR assays were performed with 100 ng of genomic bacterial DNA, 10 pmol of each primer (Mission Biotech, Taipei city, Taiwan), 10 μL of Phusion PCR Master Mix (Thermo Fisher, Waltham, MA, USA), and ultra-pure water to a final reaction volume of 20 μL. The PCR conditions included an initial denaturation at 98 °C for 1 min, followed by 30 cycles of 98 °C for 1 s, 58 °C for 5 s, and 72 °C for 10 s, with a final extension at 72 °C for 1 min. PCR amplicons were electrophoresed on a 1.5% (*w*/*v*) agarose gel in the TAE buffer. A 100 bp DNA ladder (Promega Corporation, Madison, WI, USA) was included in each run. After electrophoresis, the gels were photographed under UV at 260 nm.

### 2.6. MIC of Antibiotics

The minimum inhibitory concentration (MIC) was tested with BD Phoenix CPO panel NMIC 500 (BD) for 25 different antibiotics, including ceftazidime–avibactam and carbapenems, using true doubling dilutions. CPO with Class A/D carbapenemase is typically susceptible to ceftazidime–avibactam, whereas Class B carbapenemase is resistant to ceftazidime–avibactam. The cutoff of ≤8/4 μg/mL was used to determine susceptibility to ceftazidime–avibactam, while ≥16/4 μg/mL was interpreted as resistant to ceftazidime–avibactam.

## 3. Results

### 3.1. Discrepancy of MPCR Results Based on Different Colony-Selection Methods (The FirstAll Method versus The Conventional Method)

The MPCR results from FirstAll concurred with conventional colony-selection methods (79.8%, 379/475) (Figure 2). Amongst the discrepant cases, 53 (11.2% (53/475)) isolates showed a positive result with FirstAll but were reported as negative by the conventional method. The results indicate that 11.2% of the CPO isolates tested falsely as negative for carbapenemase by the conventional colony picking-up method.

### 3.2. Carbapenemase Testing Comparison (CPO Panel versus mCIM/eCIM) Based on FirstAll

For most CPO species except *K. pneumoniae*, the CPO panel results agreed with mCIM/eCIM well (Figure 3). The CPO panel also provided the exact classifications (i.e., Class A or D, Class B) for most CPO species. In the case of mCIM(+)/eCIM(−) *K. pneumoniae* isolates, nearly half the isolates (43/87) were from Class A or D. However, for the other half of *K. pneumoniae* isolates with mCIM(+)/eCIM(−), the class was not determined. Similarly, in *K. pneumoniae* isolates with mCIM(+)/eCIM(+), 38.60% (22/57) were Class A or D, while only 13.56% (8/57) were Class B. Of note, 47.37% (27/57) were *K. pneumoniae* isolates with mCIM(+)/eCIM(+), carbapenemase was detected by the CPO panel, but the class was not determined.

### 3.3. Carbapenemase Testing Comparison (CPO Panel versus MPCR) Based on FirstAll

The phenotypic results of the CPO panel were also compared with the genotypic results by using PCR as the reference method. The comparisons in different species were joined together to avoid scattered results (Figure 4A). For the “Class A or D” results from the CPO panel, 70.5% (79/112) was in line with MPCR results, while 2.68% (3/112) was IMP/NDM/VIM, 18.75% (21/112) was categorized as “Others” (i.e., multiple genes detected), and 6.25% (7/112) was negative. For the “Class B” category, 60.8% (48/79) agreed with MPCR, but notably, 19.0% (15/79) were called negative. With the FirstAll method adopted, a considerable proportion (18.7%, 89/475) was determined to be positive for carbapenemase by the CPO panel but failed to be specifically categorized (i.e., the “Class Unknown”). In cases defined as negative by the CPO panel, the majority (65.0%, 128/197) was also negative by MPCR. However, 35.0% of isolates were still found to harbor the carbapenemase genes.

### 3.4. Carbapenemase Testing Comparison (Carba5 versus MPCR) Based on FirstAll

The performance of Carba5 was evaluated using MPCR as the reference method. A concordant rate of around 73% was noted for all the categories (Figure 4B). Specifically, 78.8% (119/151) of cases with a KPC/OXA result showed on the Carba5 test as well as the MPCR. For the IMP/NDM/VIM class, 82.4% (61/74) of IMP/NDM/VIM cases tested by Carba5 revealed the same pattern on MPCR. For all the cases called negative for carbapenemase by Carba5, 66.7% (154/231) cases also showed negative results in MPCR. Of note, about 10% of cases that were determined as a specific class by Carba5 were classified as “Others” by MPCR (for class KPC/OXA: 16/151; for class IMP/NDM/VIM: 9/74), indicating the ability for MPCR to detect more than one carbapenemase classes in a sample.

### 3.5. Carbapenemase Testing Comparison (CPO Panel versus Carba5) Based on FirstAll

The CPO panel results generally agreed with the Carba5 results (Figure 5A) except for the “Class Unknown” category. Specifically, 85.5% (94/110) “Class A or D”, 67.1% (53/79) “Class B”, and 94.9% (187/197) “Negative” results called by the CPO panel were also in line with the results by using Carba5. Notably, the agreement between the two methods was especially low for Class B carbapenemase. Moreover, a considerable proportion (19.0%, 44/231) of Carba5-negative isolates were read as positive by the CPO panel. Further comparison was carried out for the MPCR-positive isolates (Figure 5B). Amongst the isolates that were considered carbapenemase-positive by MPCR (the reference method), result agreements between the CPO panel and the Carba5 in “Class A or D” and “Class B” were 88.3% (91/103) and 79.7% (51/64), respectively. Generally, the agreements between the CPO panel and the Carba5 were similar in either the MPCR-positive subgroup or in the whole population. Of note, there are approximately 19.8% (60/303) of MPCR-positive cases that were not detected by either the CPO panel or the Carba5.

### 3.6. Association between Carbapenemase Tests and MICs of Ceftazidime–Avibactam

Ceftazidime–avibactam can be used to treat CPOs with Class A/D carbapenemase but not for Class B carbapenemase. Thus, an agreement between the carbapenemase test and ceftazidime–avibactam MIC is important for clinical decisions. For “Class A or D”, as tested by the CPO panel, 92.7% (102/110) of the cases were susceptible to ceftazidime–avibactam (Figure 6A). For “Class B”, also tested by the CPO panel, 75.9% (60/79) of the cases were resistant to ceftazidime–avibactam, but approximately 24.1% (19/79) of the cases were interpreted as susceptible. Of note, for the “Class Unknown” category, a biurnal MIC distribution was found, as follows: 73.0% (65/89) of the cases were susceptible to ceftazidime–avibactam, while 27.0% (24/89) of the strains were resistant to ceftazidime–avibactam. A similar pattern of agreement can be observed between Carba5 and ceftazidime–avibactam MIC (Figure 6B). On top of that, the Carba5 also showed a higher agreement with ceftazidime–avibactam MIC, as follows: 94.7% (143/151) of the “KPC/OXA” cases were susceptible to ceftazidime–avibactam, while 86.5% (64/74) of the “IMP/NDM/VIM” cases were resistant to ceftazidime–avibactam. For the “Others” category, multiple carbapenemase classes were identified, and most of them (68.4% (13/19)) presented as resistant to ceftazidime–avibactam.

## 4. Discussion

CPOs pose a significant threat to public health due to their ability to break down carbapenem antibiotics, which are often considered the last line of defense against serious infections. In this study, we performed a comparison between various carbapenemase tests utilizing an unbiased colony-selection approach, specifically the FirstAll method. Based on results from the FirstAll method, the conventional carbapenemase tests (either the Carba5 or the CPO panel) showed suboptimal diagnostic performance on carbapenemase classification.

The fundamental assumption in clinical microbiology laboratories that colonies on an agar plate with the same morphology are considered identical has been challenged and discussed in recent years. Specifically, the organisms that were found in the bloodstream infections of CRKP (CRKP-BSI) do not meet this assumption [13]. Gastrointestinal colonization with *K. pneumoniae* is a major predisposing risk factor for infection and a hub for the dispersal of resistance [14,17]. This risk factor is considered endogenous rather than exogenous. The KP strains usually colonize the host’s gut for a long time [14], thus meaning that those KP strains typically exhibit multiple and diverse antibiotic resistance mechanisms for survival against antibiotic challenges on numerous occasions [14,18,19]. The KP strains likely reach a colonization equilibrium, and there is no survival advantage for one KP strain over another. Once the host’s immunity is weakened, the diverse KP strains will disseminate out from the gut and enter the bloodstream. In this case, there is likely a considerable amount of strain diversity within a specimen, even if the morphologies of the strains are identical.

To avoid any selection bias in lab tests, we developed the FirstAll method to increase the detection of strain diversity on colony selection. The results showed that around 10% of the CPO strains were found to be positive in FirstAll but were negative using the conventional method (Figure 2). The 10% discordance between the colony-selection methods indicates that the conventional colony-selection method is insufficient for the detection of intra-specimen diversity. The inability to detect the diverse strains could lead to incorrect antibiotic selection and eventual unfavorable clinical outcomes. Our findings posed that the FirstAll method would be a more comprehensive, unbiased, and adequate colony-selection method. By contrast, the utility of the conventional colony-selection method needs to be further investigated to avoid false negatives of carbapenemase detection, as noted in our study.

Intra-specimen strain diversity would account for the discrepancy between various carbapenemase tests. In the comparison study between the CPO panel and mCIM/eCIM, we showed the subgroup comparisons for all the bacterial species (Figure 3). Generally, the CPO panel results aligned well with the mCIM/eCIM results. Interestingly, KP was the only bacterial species that the CPO panel results did not agree with as much as the results from mCIM/eCIM, highlighting the species-specific nature of these tests. The CPO panel did not allocate specific classes for approximately 50% of KP species. The high occurrence of cases classified as “Class Unknown”—meaning carbapenemase-positive cases whose class could not be determined—is noted in the KP species, in comparison with any other species. The high percentage of “Class Unknown” in KP strains implies that there were diverse carbapenemase classes, which were detected and identified by using FirstAll as the colony-selection method. While other bacterial species did not show a high percentage of “Class Unknown”, the underlying mechanism between the bacterial species is worthy of further investigation.

With MPCR as the reference method, both the CPO panel and the Carba5 showed a high concurrence with MPCR when specific classes were allocated (Figure 4). The major difference between the CPO panel and the Carba5 arises from strains, as multiple carbapenemases classified by the Carba5 were classified as “Class Unknown” by the CPO panel. The information about specific classes is beneficial when choosing the correct antibiotic (e.g., ceftazidime–avibactam). For the MPCR-positive subgroup, 19.8% of the cases were neither reported in the CPO panel nor in the Carba5. Evaluating results from the FirstAll colony-selection method, the diagnostic performance of the CPO panel or the Carba5 was lower than that reported in previous studies [20,21]. The hypothesis is that the FirstAll method unbiasedly collects more colonies than the conventional colony-collection method, and in some cases, it is able to catch the minor variants even though they may only account for a small portion of the organisms detected. The minority variants may be detected by higher sensitivity methods (such as MPCR), but not by phenotypic assays. The clinical impacts of extra carbapenemase detection using the unbiased colony-selection method will be an interesting topic to investigate.

Several limitations should be addressed in this study. First, the CPO strains were isolated from a single referral medical center in East Asia. Given the fact that local circulating strains may differ considerably between different areas, hospital types, and local epidemiology, those factors should be taken into consideration in interpreting the results. Second, genomic information was not tested, especially for the cases showing multiple different types of carbapenemases. It is not clear whether those variations in carbapenemases derived from a single strain or from multiple strains that were collected by the FirstAll method. A genomic study, preferably long-read sequencing, would be necessary to reveal the genetic context between the carbapenemase genes. *K. pneumoniae* accounts for the majority of the CPOs in the study as well as in a real-world setting. The comparisons would be largely dominated by *K. pneumoniae* over other CPO species due to its high abundance. While including the other minor CPO species increased the inclusivity of the study, collecting more isolates for the minor CPO species is warranted to investigate how the carbapenem tests would perform with an unbiased colony-selection method. Furthermore, the findings from this study emphasize the complexity and the potential clinical impacts of the carbapenemase tests.

## 5. Conclusions

This study demonstrated that the use of an unbiased colony-selection method (i.e., FirstAll) significantly improved the detection of CPOs compared to conventional colony-selection methods. The findings revealed that approximately 11.2% of CPO isolates were falsely negative when conventional methods were used, highlighting the advantage of the FirstAll method in reducing diagnostic errors. However, this improvement in detection comes at a cost, as the diagnostic performance of both the Carba5 and the CPO panel was compromised when using the FirstAll method, with sensitivities of 74.6% and 77.2%, respectively. Despite this, the Carba5 test provided accurate class-specific carbapenemase detection, which is crucial for guiding clinical decision-making. The study also confirmed that ceftazidime–avibactam use, according to the results from either the FirstAll-based Carba5 test or the CPO panel, remains suitable, particularly given the high concordance rates with MICs for Class A/D carbapenemases. The study showed the importance of selecting an appropriate colony-selection method on both the accuracy of carbapenemase detection and the subsequent therapeutic strategies.

## Figures and Tables

**Figure 1 biomedicines-12-02134-f001:**
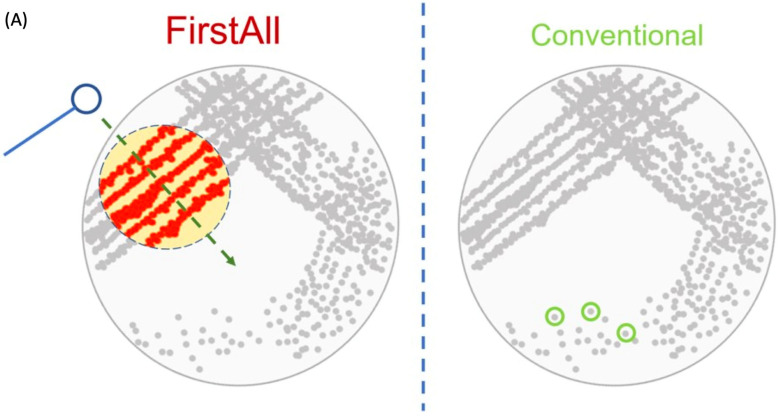
(**A**) **Schematic illustration of FirstAll method for unbiased colony selection.** We propose the FirstAll method to increase bacterial diversity and avoid selection bias prior to analyses. In the FirstAll method, we scratch on the agar with a direction vertical to the streaking of the first streak area. By contrast, only several single colonies are picked up for the conventional method. (**B**) **Illustrative figure of the study.** There are three stages in the study: (1) comparison between FirstAll and single colony-collection method by using MPCR; (2) comparisons between different carbapenemase tests based on FirstAll; (3) association between carbapenemase tests and MICs of ceftazidime–avibactam.

**Figure 2 biomedicines-12-02134-f002:**
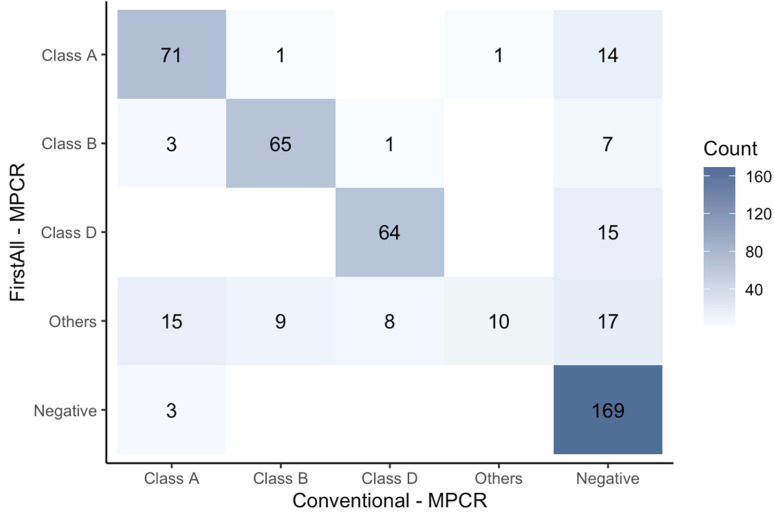
**Comparison between FirstAll and conventional colony-selection methods.** Most of the CPO isolates show concordant MPCR results with both the FirstAll and conventional colony-selection methods. Of note, 53 out of 475 isolates reveal MPCR as positive for the FirstAll method but negative for the conventional method, indicating FirstAll would be an unbiased colony-selection method.

**Figure 3 biomedicines-12-02134-f003:**
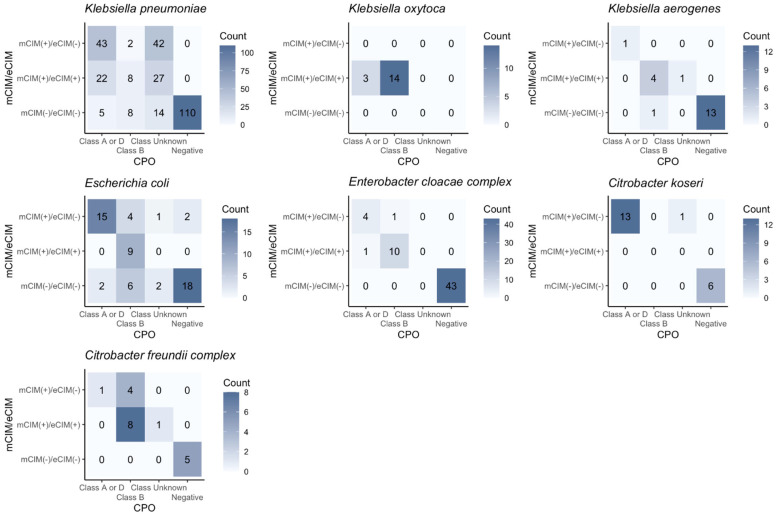
**Comparison between CPO panel and mCIM/eCIM.** General categorical agreements between the CPO panel and mCIM/eCIM can be found along the diagonal lines except in K. pneumoniae. The CPO panel can detect the existence of carbapenemase in K. pneumoniae, but the class cannot be determined. Specifically, around half of K. pneumoniae isolates (42/87) with mCIM(+)/eCIM(−) (i.e., regarded as Class A/D carbapenemase) are categorized as “Class Unknown”; 27 out of 57 K. pneumoniae isolates with mCIM(+)/eCIM(+) (i.e., regarded as Class B carbapenemase) are categorized as “Class Unknown”.

**Figure 4 biomedicines-12-02134-f004:**
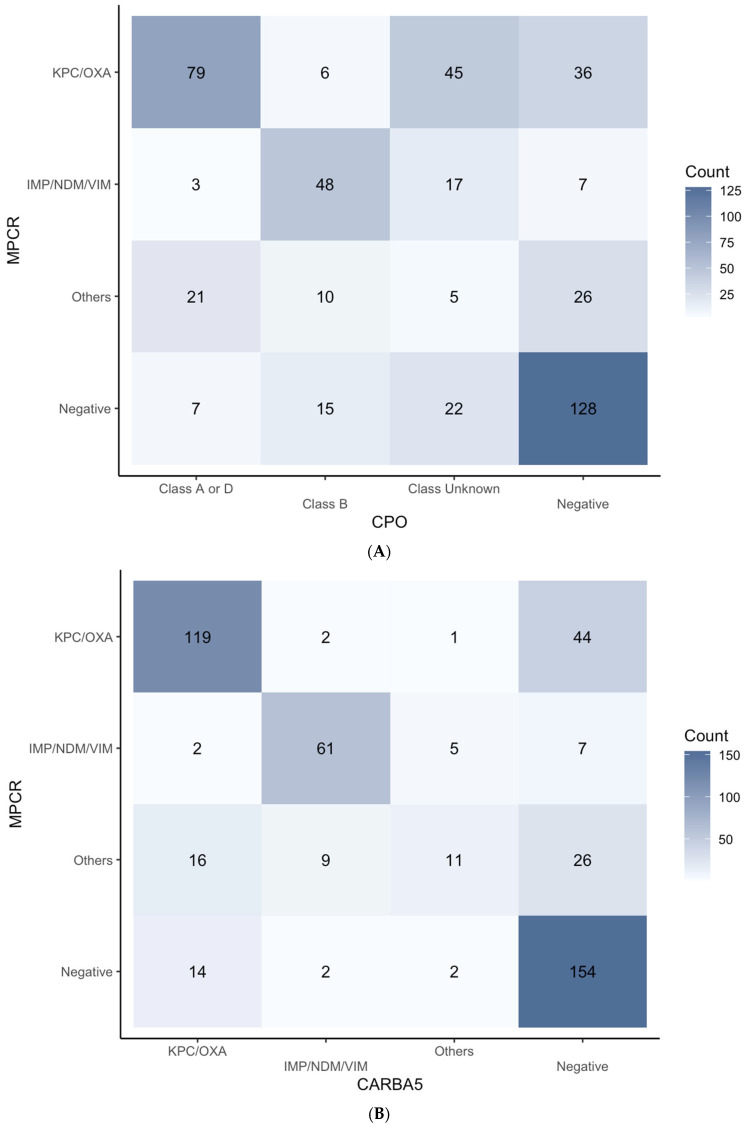
(**A**) **Comparison between CPO panel and MPCR.** The CPO panel and MPCR results are generally in agreement except for the “Class Unknown” of the CPO panel. The “Class Unknown” accounts for 18.7% [89/475] of all the isolates. The overall sensitivity and specificity are 77.2% [234/303] and 74.4% [128/172], respectively. (**B**) **Comparison between Carba5 and MPCR.** The agreement in results between Carba5 and MPCR is high for all the categories. The overall sensitivity and specificity are 74.6% [226/303] and 89.5% [154/172], respectively.

**Figure 5 biomedicines-12-02134-f005:**
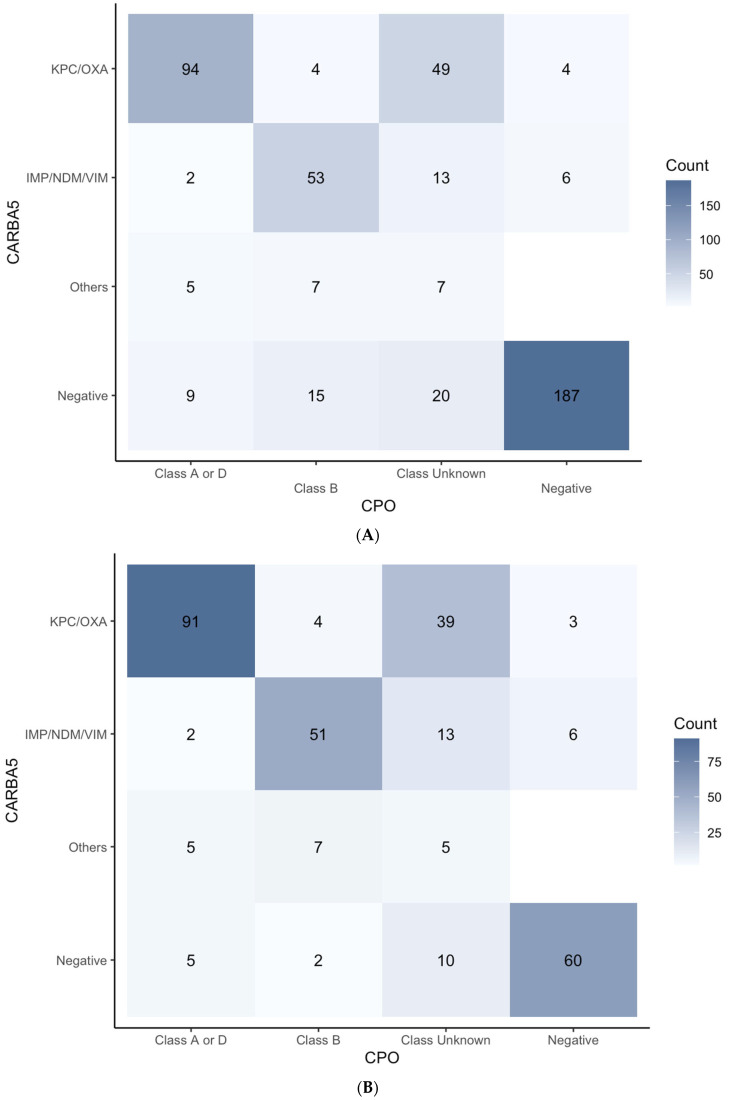
(**A**) **Comparison between CPO panel and Carba5.** The agreement between the two methods is generally congruent for the “Class A or D” and “Negative” categories, but the agreement is lower for the “Class B” category. Moreover, for the isolates that were read as positive by the CPO panel, 34.25% (100/292) of those are positive for carbapenemase but class unknown. (**B**) **Comparison between CPO panel and Carba5 on the MPCR-positive strains**.

**Figure 6 biomedicines-12-02134-f006:**
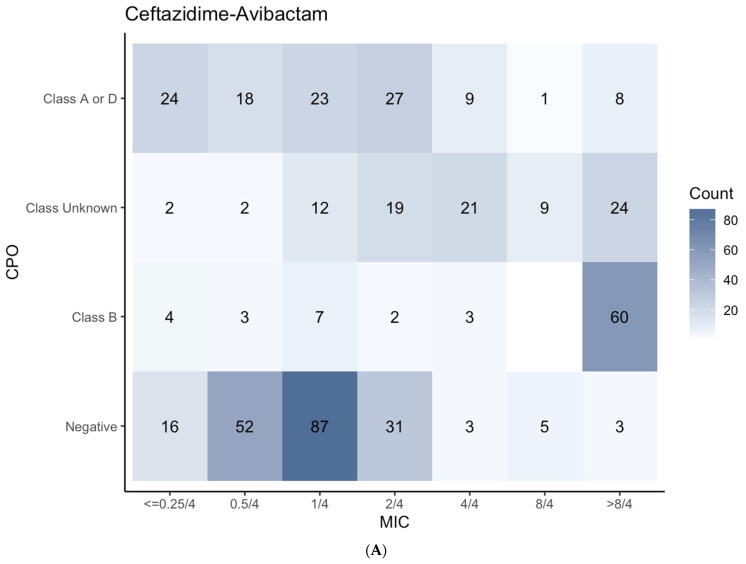
(**A**) MICs of ceftazidime–avibactam in different classes of the CPO panel. (**B**) MICs of ceftazidime–avibactam in different classes of the Carba5.

**Table 1 biomedicines-12-02134-t001:** **Primers sequence for top 5 carbapenemase genes.** We designed primer sets for the five most common carbapenemase genes (i.e., KPC, OXA, VIM, NDM, IMP). The size of the expected amplicons was designed to be different so that the amplicons could be distinguishable in a multiplex PCR setting. F: forward; R: reverse.

Gene	Primer	Nucleotide Sequence (5′-3′)	Amplicon Size (bp)
KPC	F R	CTGACCAACCTCGTCGCGGAAC TTGTTAGGCGCCCGGGTGTAGA	731
OXA	F R	TGGGATGGACAGACGCGCGATA CCAACCGACCCACCAGCCAATC	393
VIM	F R	TTGGACTTCCCGTAACGCGTGC AGCTCTACTGGACCGAAGCGCA	208
NDM	F R	AAGGCCAAGTCGCTCGGCAATC ACTCGTCGCAAAGCCCAGCTTC	264
IMP	F R	GCAGGAGCGGCTTTGCCTGATT GGCGGACTTTGGCCAAGCTTCT	543

## Data Availability

The data presented in this study are available on request from the corresponding author due to restrictions.

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
