# Peer review of "Comparison of Multiple Carbapenemase Tests Based on an Unbiased Colony-Selection Method"

_biomedicines, 2024, doi:10.3390/biomedicines12092134_

Round 1

Reviewer 1 Report

Comments and Suggestions for Authors

Overall, this study has some good points, but there are also some serious logical errors that need to be addressed.

For example, when comparing the results of carbapenemase detection using the mCIM method between two Klebsiella pneumoniae groups (FirstAll and single-colony), it would be necessary to include the results of PCR analysis as a control to accurately compare the two colony collection methods. Without this information, it is not possible to determine which collection method allows for more accurate detection of carbapenemase production in a particular strain. The authors have not provided any information on this, so a comparison between the two methods cannot be made. However, it is possible to make, as the researchers have already conducted PCR testing on all the strains under study based on the data presented.

Therefore, the authors could provide a chart showing the results of PCR analysis for all strains studied, including both matched and unmatched samples for both groups. This would allow them to easily compare the results and calculate the percentage of accuracy for each colony picking method. With this information, they could then make an informed decision about which method is most effective for detecting carbapenemase resistance in Klebsiella pneumoniae.

The second important point is that it is not very meaningful to compare the results of carbapenemase tests conducted using only the “FirstAll” colony collection method. It's unclear exactly what this comparison is supposed to reveal. The fact that test results between different tests don't match? However, this does not mean that we can draw any conclusions yet, as the reliability of the “FirstAll” colony collection method has not been yet established (as mentioned in the previous paragraph). Therefore, we do not know why the results of the carbapenemase detection tests differ from each other in this way. The percentage of false positive and false negative results can be influenced by several factors, including the colony selection method, the specific test used, and the bacterial species. In order to make an informed decision about the significance of using a particular approach, it is essential to thoroughly investigate all these factors before drawing any conclusions.

 As for the design of the study, comparison of different carbapenemase detection tests should be addressed first when using a conventional single-colony method, so it might be useful for readers. Then, conduct the same tests using the “FirstAll” method, and compare all these results with those obtained using the control PCR method. Only then will the significance of all this become clear. Then we can see, for example, that the "FirstAll" method more frequently allowed us to detect carbapenemases in strains that weren't identified using the single-colony method, as well as determine the most reliable carbapenemase detection test. If the comparison of the results of testing for carbapenemases is conducted using only the "First All" colony collection method, the impact of which on the results and accuracy is unknown and has not been established in this study (as I mentioned in the previous paragraph), then it is probably not advisable to do so. I would suggest either conducting additional research using a traditional method to collect colonies and make a fair comparison with the control (PCR) in order to support this part of your research, or removing it from your paper entirely.

Introduction, Lines 108-112:

“To begin with, the discrepancy between the FirstAll method and the conventional colony-picking method was ascertained by comparing the FirstAll method with the common carbapenemase tests for seven bacterial species. This study comprehensively analyses the performance of various carbapenemase tests and contribute to our understanding of within-host bacterial heterogeneity and its impact on diagnostic methods.”

As it stands, the purpose of the study appears to be incorrect. The problem lies in the fact that there is a lack of clarity about what was actually compared to what. I would like to suggest rephrasing this in a different way:

The aim of the study was to compare the results of carbapenemase detection by mCIM test in Klebsiella pneumoniae strains using two different colony collection methods: FirsAll and the conventional method, as well as to compare the performance of several carbapenemase detection tests using the FirstAll method for colony collection with seven Enterobacteriaceae species.

One more thing to note, the number of strains that underwent carbapenemase detection testing (shown in figures 2, 4, and 5) is different from each other and also differs from the number of strains mentioned in the materials and methods section.

Author Response

Dear reviewer,

Please refer to the response letter attached.

HsinYao Wang

Reviewer 2 Report

Comments and Suggestions for Authors

The purpose of the study is unclear: is it to demonstrate the better applicability of the FirstAll method compared to the conventional colony-sorting method, or is it to compare the different diagnostic methods?

I find the work rather confusing. 

The method proposed by the

authors, FirstAll, does not seem so new. It is already used in diagnostic laboratories when one is certain that only one micro-organism is present in the culture, and therefore it is pure. Therefore, I would avoid calling it a 'new method'. I would suggest saying that it is a method already in use and focus on ascertaining the purity of the strains analysed.

Nevertheless, this method has the potential to be valid, but to prove its true validity, it is essential that it be tested on bacterial strains of different species.  There is no scientific evidence to assume that these results can be generalised to the entire Enterobacteriaceae family. 

In my opinion, authors should test more bacterial species with the two methods to demonstrate the true potential of the FirstAll method.

Since the authors have additional bacterial species I would invite them to use them to compare methods.

The presentation of the results is confusing. 

In my opinion, if the aim of the authors is to recommend the flirt method for therapeutic purposes, the two phases of the work should be divided: I would recommend investigating the difference between the two methods (FirstAll method and the traditional method) with other strains, and only then focus on comparing all other methods.

I am really sorry because the authors have done a considerable amount of work, but I do not recommend publishing this manuscript in this form.

Author Response

(The authors gave the same response as above.)

Reviewer 3 Report

Comments and Suggestions for Authors

The manuscript "Comparison of Multiple Carbapenemase Tests Based on an Unbiased Colony Selection Method" explored the impact of an unbiased colony selection on carbapenemase testing and assessed its relevance on various tests. Under the unbiased colony selection method, and diagnostic performance of either the Carba5 or the CPO panel was compromised and lower sensitivity. Class-specific detection by the Carba5 would aid clinical decision-making. Ceftazidime-avibactam use based on results from either the CPO panel or the Carba5 is suitable.

Generally, the manuscript is improved a lot and can be published in „Biomedicines“, However, I still have minor comments:

1)   Figures, most of the Figures are not clear, What about designing tables to be more clear for the readers?

2) What about the statistical analysis?

3) There is a very limitation of cited references. authors cited only 17 references. 

4) Other minor comments, please see the attached file.

Comments on the Quality of English Language

Minor editing of English language required

Author Response

(The authors gave the same response as above.)

Round 2

Reviewer 1 Report

Comments and Suggestions for Authors

I would like to express my gratitude for the authors' efforts to incorporate my comments into their article. The adjustments they have made are satisfactory and I believe that the article now provides a more accurate representation of the differences in test results based on the method of sample collection. However, I would appreciate it if the conclusions could be further elaborated based on the findings from the study.

Author Response

Dear Reviewer,

Reviewer #1

I would like to express my gratitude for the authors' efforts to incorporate my comments into their article. The adjustments they have made are satisfactory and I believe that the article now provides a more accurate representation of the differences in test results based on the method of sample collection. However, I would appreciate it if the conclusions could be further elaborated based on the findings from the study.

Response:

Thank you for the suggestion. We have expanded and elaborated our conclusions based on the findings from the study, as follows (Page 17, Line 404-417):

The study demonstrated that the use of an unbiased colony selection method (i.e. FirstAll) significantly improved the detection of CPOs compared to conventional colony selection methods. The findings revealed that approximately 11.2% of CPO isolates were falsely negative when conventional methods were used, highlighting the advantage of the FirstAll method in reducing diagnostic errors. However, this improvement in detection comes at a cost, as the diagnostic performance of both the Carba5 and the CPO panel was compromised when using the FirstAll method, with sensitivities of 74.6% and 77.2%, respectively. Despite this, the Carba5 test provided accurate class-specific carbapenemase detection, which is crucial for guiding clinical decision-making. The study also confirmed that ceftazidime-avibactam use according to the results from either FirstAll-based Carba5 test or the CPO panel remains suitable, particularly given the high concordance rates with MICs for class A/D carbapenemases. The study showed the importance of selecting an appropriate colony selection method on both the accuracy of carbapenemase detection and the subsequent therapeutic strategies.

Reviewer 3 Report

Comments and Suggestions for Authors

The manuscript is improved a lot and could be published. However, there is a problem with references and citation styles. See lines 307-320. "(13)[10.51585/gjm.2021.0004]. This risk factor is 312 considered endogenous rather than exogenous. The KP strains usually colonize the host’ 313 gut for a long time (13), thus meaning that those KP strains typically exhibit multiple and 314 diverse antibiotic resistance mechanisms for survival against antibiotics challenges in nu- 315 merous occasions (13)[10.1186/s13567-020-00875-w][10.1128/mSphere.00734-21]." 

Comments on the Quality of English Language

The manuscript is improved a lot. 

Author Response

Reviewer #3

The manuscript is improved a lot and could be published. However, there is a problem with references and citation styles. See lines 307-320. "(13)[10.51585/gjm.2021.0004]. This risk factor is 312 considered endogenous rather than exogenous. The KP strains usually colonize the host’ 313 gut for a long time (13), thus meaning that those KP strains typically exhibit multiple and 314 diverse antibiotic resistance mechanisms for survival against antibiotics challenges in nu- 315 merous occasions (13)[10.1186/s13567-020-00875-w][10.1128/mSphere.00734-21]."

Response:

Thank you for reminding us of the citation styles. The citations were listed with their DOI information only because citation managing function had been inactivated in the manuscript. We will be asking editorial office to help us incorporate the newly added citations into the revised manuscript with the correct style.